# Genomic Evaluation of Harvest Weight Uniformity in *Penaeus vannamei* Under a 3FAM Design Incorporating Indirect Genetic Effect

**DOI:** 10.3390/biology14040328

**Published:** 2025-03-24

**Authors:** Siqi Gao, Yan Xia, Jie Kong, Xianhong Meng, Kun Luo, Juan Sui, Ping Dai, Jian Tan, Xupeng Li, Jiawang Cao, Baolong Chen, Qiang Fu, Qun Xing, Yi Tian, Junyu Liu, Sheng Luan

**Affiliations:** 1College of Fisheries and Life Science, Dalian Ocean University, Dalian 116023, China; gaosq1216@163.com (S.G.); tianyi@dlou.edu.cn (Y.T.); 2State Key Laboratory of Mariculture Biobreeding and Sustainable Goods, Yellow Sea Fisheries Research Institute, Chinese Academy of Fishery Sciences, Qingdao 266071, China; xiayan1207@126.com (Y.X.); kongjie@ysfri.ac.cn (J.K.); mengxianhong@ysfri.ac.cn (X.M.); luokun@ysfri.ac.cn (K.L.); suijuan0313@126.com (J.S.); daiping54@163.com (P.D.); tanjian@163.com (J.T.); lixupeng@ysfri.ac.cn (X.L.); caojw@ysfri.ac.cn (J.C.); chenbl@ysfri.ac.cn (B.C.); oucfuq@163.com (Q.F.); 3Laboratory for Marine Fisheries Science and Food Production Processes, Qingdao National Laboratory for Marine Science and Technology, Qingdao 266235, China; 4BLUP Aquabreed Co., Ltd., Weifang 261311, China; xingqun527@163.com

**Keywords:** *Penaeus vannamei*, harvest weight uniformity, indirect genetic effects, H matrix, genetic parameters

## Abstract

The Pacific white shrimp (*Penaeus vannamei*) is a key species in global aquaculture. However, significant variation in harvest weight—sometimes exceeding a tenfold difference among full-sibling individuals—poses challenges to farm productivity and profitability. A crucial first step in improving this trait through selective breeding is the precise estimation of genetic parameters. While previous studies have reported such variability in various aquaculture species, including shrimp, existing assessment methods require further refinement. It is essential to account for key environmental factors, such as inter-individual interaction effects, which can influence both individual productivity and welfare. Additionally, incorporating genomic information rather than relying solely on pedigree data can provide a clearer representation of genetic relationships among individuals. These studies have aimed to enhance the accuracy of genetic variance estimation by improving models for harvest weight uniformity, ultimately assessing the feasibility of including this trait in selective breeding programs. This study employed a grouping design with three families per group involving 40 families of shrimps containing 36 shrimps per family to estimate the contribution of direct and indirect genetic effects on harvest weight uniformity. This was estimated using a hierarchical generalized linear model, integrating genomic and indirect genetic effect data. Our findings indicate that integrating social interaction effects and genomic information has the potential to improve the precision of genetic evaluations and advancing breeding strategies for enhanced productivity in *P. vannamei*.

## 1. Introduction

The Pacific white shrimp (*Penaeus vannamei*) is a cornerstone species in global aquaculture due to its exceptional adaptability, rapid growth rate, and high economic value. In 2022, global production of *P. vannamei* reached 6.8 million tons, cementing its position as the leading aquaculture species in terms of both production volume and economic significance [1]. However, substantial within-family variation in harvest weight (HW), often exceeding tenfold among full-sib individuals, persists as a major challenge [2,3]. This variability undermines production efficiency and economic returns, underscoring the critical need to improve the harvest weight uniformity (HWU). HWU is defined as the degree of consistency in HW among individuals within a population, often quantified through the genetic variance in residual variance [4,5]. Lower residual variance indicates greater uniformity, which is desirable for optimizing production efficiency and economic value [6,7]. Economic analyses have demonstrated that incorporating HWU into breeding objectives can increase selection response by over 40% in pigs [8]. Although, considering HWU is very important in the selective breeding of *P. vannamei*, progress has been limited by the absence of robust experimental design, evaluation methods, and statistical models capable of simultaneously addressing trait means and residual variance [9].

The double hierarchical generalized linear model (DHGLM), based on restricted maximum likelihood (REML), is the predominant statistical approach for the genetic evaluation of uniformity traits [10]. This model consists of the following two interconnected components: a mean model that estimates genetic and environmental effects on the mean trait, and a residual variance model that evaluates genetic heterogeneity in residual variance [10,11]. The residual variance model uses the squared residuals from the mean model as the dependent variable, enabling the genetic analysis of environmental sensitivity and stability among individuals within a population [12]. By modeling residual variance as a function of genetic and environmental effects, DHGLM provides insights into the genetic control of variability, offering a comprehensive framework for studying genotype–environment interactions [6,13]. These capabilities are particularly valuable in breeding programs aimed at improving uniformity, as they allow for the integration of residual variance into genetic evaluations and the identification of individuals or lines with enhanced environmental robustness [6,14]. Studies on uniformity traits have been conducted across livestock and aquatic species, including pigs, cattle, rainbow trout, and Atlantic salmon [13,14,15,16]. In *P. vannamei*, Garcia et al. [2] used sire–dam DHGLM to estimate (co)variance components for HW and HWU based on 149,919 and 164,023 records of HW and survival in three environments with different densities. The results showed a substantial genetic variation in uniformity. HW and HWU exhibited different genetic correlations with survival, which were influenced by the production system and environmental conditions. However, existing assessment methods need further refinement to precisely estimate genetic parameters, such as capturing genetic variation associated with social interactions and integrating genomic information. This would lead to more accurate breeding values for uniformity, thereby accelerating the breeding progress of HWU of *P. vannamei*.

Traditional quantitative genetic models primarily consider an individual’s phenotype as the sum of its direct genetic effects (DGEs) and environmental effects [17]. However, environmental effects also encompass components of the social environment, where competition and interactions among individuals can significantly influence traits such as growth rate, feed intake, and mortality [18,19,20,21]. These social interactions are known to contribute to variability in HW [22], as seen in species such as Nile tilapia [23], Sunfish [24], and Pacific white shrimp [25]. Consequently, social interactions are an important determinant of HWU. To capture the genetic variation associated with these interactions, indirect genetic effects (IGEs) have been incorporated into genetic evaluations. By accounting for the influence of social interactions, IGEs provide a more comprehensive understanding of the factors affecting HWU [26]. A grouping design with three families per group (3FAM), where each test group consists of three families and evaluates each family across three independent groups, is particularly effective for analyzing IGEs. By increasing the number of test groups per family, this design improves the ability to estimate variance components for HWU, particularly for traits with low heritability. The structure reduces confounding between direct and social genetic effects by enabling multiple independent tests of each family across groups [27,28,29].

Given the low heritability of uniformity traits, genomic selection (GS) has become essential for improving selection accuracy [16]. It enables the estimation of breeding values by estimating the effects of markers at a high density covering the whole genome, thus being able to explain greater genetic variations in given traits. The single-step genomic best linear unbiased prediction (ssGBLUP) method, introduced by Legarra et al. [30], combines genomic, pedigree, and phenotypic information into a unified H matrix for breeding value estimation. This approach has consistently demonstrated improvements in selection accuracy across various aquatic species, including Atlantic salmon, rainbow trout, Nile tilapia, and channel catfish [16,31,32,33]. For instance, in *Macrobrachium rosenbergii*, ssGBLUP increased predictive accuracy by 16.67–42.42% compared to pedigree-based BLUP (pBLUP), with a remarkable 143.95% gain in genetic improvement [34]. However, the application of ssGBLUP to uniformity traits in aquaculture remains limited. Sae-Lim et al. [16] reported a modest but significant improvement of 1.3–13.9% in predictive accuracy for body weight uniformity in Atlantic salmon when using ssGBLUP over pBLUP.

This study aimed to estimate the variance components for HWU in *P. vannamei* using the DHGLM framework, which integrates IGEs into the mean model. Additionally, it sought to evaluate the predictive accuracy of GS for HWU using a 3FAM experimental design involving 1440 individuals from 40 families. The practical application of GS in shrimp breeding was assessed by examining its potential to improve uniformity traits under typical breeding program conditions.

## 2. Materials and Methods

### 2.1. Samples and Experimental Design

This study utilized a 3FAM experimental design to evaluate the effect of incorporating IGEs on genetic parameters related to HWU in *P. vannamei*. The experimental shrimp were derived from 40 families of a nucleus breeding population, spawned in May 2022, and provided by BLUP Aquabreed Co., Ltd., Changyi City, Weifang, Shandong Province, China. Once the shrimp reached the P60 size specification of approximately 4.8 g, the sixth abdominal segment of each shrimp was tagged using visible implant elastomer (VIE) markers. From each family, 36 individuals were randomly selected, yielding a total of 1440 shrimp for the trial.

Graphical representation of 3FAM is shown in Figure 1. Each family was divided into three groups, with twelve shrimp per group, and allocated to three cylindrical cages for combined testing with six other families [27]. Each cage contained shrimp from three distinct families, with family combinations optimized based on kinship coefficients and trait variation coefficients. Specifically, in each cage, the coefficient of variation (CV) in body weight among the three families had to be less than or equal to the CV for body weight across 40 families. Additionally, the CV for body weight of all individuals within the cage had to be less than or equal to the CV for body weight across all tested individuals. Additionally, the average kinship coefficient among the three families in each cage had to be less than or equal to the average kinship coefficient among the 40 families. Notably, any specific pairwise family combination appeared in only one cage throughout the experiment to minimize bias in family-specific effects.

The trial lasted 55 days and was conducted in a 360 m^3^ (9 m × 40 m × 1 m) earthen pond located inside a small shed. The shed was covered with sun-shading plastic film and equipped with ventilation outlets to regulate light and airflow. Water quality parameters remained within acceptable thresholds throughout the trial, as follows: the pH ranged from 8.04 to 8.94, salinity from 26 to 31 ppt, dissolved oxygen from 5.2 to 8.7 mg/L, and total alkalinity from 118.08 to 200.72 mg/L. Temperature exhibited distinct diurnal patterns, with daytime values spanning from 21.8 °C to 34.0 °C and night-time measurements varying from 21.4 °C to 28.2 °C. Forty cylindrical cages (60 cm diameter, 80 cm height) were positioned in the pond, with 20 cages placed along each side of a central cement walkway. The cages were elevated 40 cm above the pond bottom, providing an effective water depth of 60 cm and a usable water volume of 0.17 m^3^ per cage. The stocking density was set at 212 individuals/m^3^.

At the end of the trial, data collected for each cage included survival rates, VIE marker color combinations, sex, and individual harvest weights.

### 2.2. Genotype and Quality Control

Muscle tissue samples were collected from the first abdominal segment of each shrimp and stored in 5 mL sampling tubes containing DNA preservation solution. High-purity DNA was extracted using a commercial DNA extraction kit, and a targeted sequencing library was prepared using GenoBaits probe capture technology provided by MolBreeding in Shijiazhuang, China for the “Yellow Sea Chip No. 1” series. Specifically, the 55K SNP Panel, comprising 56,214 SNP markers, was used for genotyping. Sequencing was performed on the MGISEQ-T7 platform, generating initial genotype data for 56,214 SNPs across 1275 individuals. Quality control (QC) of the genotype data was performed using PLINK1.9 software [35]. SNP markers were retained if they had a missing rate < 10% and a minor allele frequency (MAF) > 0.05. Individuals with a genotype call rate ≤ 80% were excluded. After QC, 49,748 SNP markers and 1220 individuals were retained for downstream analyses.

### 2.3. Statistical Analysis

The mean, minimum, maximum, standard deviation, and coefficient of variation of the shrimp harvest weights in 40 cages were calculated in R-4.0.2 [36]. Genetic parameters for HW and HWU were estimated using the average information REML method in the ASReml [37]. Four statistical models were used to evaluate the effects of IGEs and the type of relationship matrix on genetic parameter estimates. The models were defined as follows: A_NoIGE, which uses the pedigree-based additive relationship matrix (A) and excludes IGEs; H_NoIGE, which uses the genomic-pedigree combined relationship matrix (H) and excludes IGEs; A_IGE, which uses the A matrix and incorporates IGEs; H_IGE, which uses the H matrix and incorporates IGEs.

In this analysis, HW represents the mean performance of harvest weight, while HWU refers to the uniformity of harvest weight, quantified as the genetic variance in residual variance. The mathematical structures of these models are presented below:

Sire–dam DHGLM models without IGEs (A_NoIGE and H_NoIGE):

The models without IGEs assume no social interactions among individuals. The mean model for HW and the residual variance model for HWU are structured as follows:$$\left[\begin{array}{c} y\\ \Psi \end{array}\right]=\left[\begin{array}{cc} X & 0\\ 0 & X_{v} \end{array}\right]\left[\begin{array}{c} b\\ b_{v} \end{array}\right]+\left[\begin{array}{cc} \left(Z_{sire}+Z_{dam}\right) & 0\\ 0 & {\left(Z_{sire}+Z_{dam}\right)}_{v} \end{array}\right]\left[\begin{array}{c} a_{u}\\ a_{u_{v}} \end{array}\right]+\left[\begin{array}{cc} Z_{t} & 0\\ 0 & 0 \end{array}\right]\left[\begin{array}{c} t\\ 0 \end{array}\right]+\left[\begin{array}{c} e\\ e_{v} \end{array}\right]$$

Sire–dam DHGLM model with IGEs (A_IGE and H_IGE):

For models incorporating IGEs, the mean model for HW includes the effect of social interactions among individuals. The models are structured as follows:$$\left[\begin{array}{c} y\\ \Psi \end{array}\right]=\left[\begin{array}{cc} X & 0\\ 0 & X_{v} \end{array}\right]\left[\begin{array}{c} b\\ b_{v} \end{array}\right]+\left[\begin{array}{cc} \left(Z_{sire}+Z_{dam}\right) & 0\\ 0 & {\left(Z_{sire}+Z_{dam}\right)}_{v} \end{array}\right]\left[\begin{array}{c} a_{u}\\ a_{u_{v}} \end{array}\right]+\left[\begin{array}{cc} Z_{s} & 0\\ 0 & 0 \end{array}\right]\left[\begin{array}{c} a_{s}\\ 0 \end{array}\right]+\left[\begin{array}{cc} Z_{t} & 0\\ 0 & 0 \end{array}\right]\left[\begin{array}{c} t\\ 0 \end{array}\right]+\left[\begin{array}{c} e\\ e_{v} \end{array}\right]$$
where y is the vector of response variables for the mean model, representing HW; Ψ is the vector of response variables for the residual variance model, representing HWU, where $\psi_{i}=\log\left({\hat{\sigma }}_{e_{i}}^{2}\right)+\frac{\frac{{\hat{e}}_{i}^{2}}{1-h_{i}}-{\hat{\sigma }}_{e_{i}}^{2}}{{\hat{\sigma }}_{e_{i}}^{2}}$, which was linearized using a Taylor series approximation in ASReml, where ${\hat{e}}_{i}^{2}$ is the squared residual of the $i^{th}$ HW records, $h_{i}$ is the diagonal element in the hat-matrix of y, and ${\hat{\sigma }}_{e_{i}}^{2}$ is the estimated residual variance in the $i^{th}$ observation in the previous iteration of ASReml; X and $X_{v}$ are the design matrix of fixed effects, where X includes sex and the interaction between sex and initial body weight at tagging for HW, while $X_{v}$ includes only sex for HWU; b and $b_{v}$ are the solutions for fixed effects in the mean and residual variance models, respectively; $Z_{sire}$ and $Z_{dam}$ are the design matrix for DGEs of sire and dam, respectively; $a_{u}$ and $a_{u_{v}}$ are the vector of DGEs of sire–dam for HW and HWU, respectively, which are assumed to follow a multivariate normal distribution for the A matrix or H matrix: $\left[\begin{array}{c} a_{u}\\ a_{u_{v}} \end{array}\right]\sim N\left(\left[\begin{array}{c} 0\\ 0 \end{array}\right],\frac{1}{4}\left[\begin{array}{cc} \sigma_{a_{d}}^{2} & \sigma_{a_{d},a_{v,exp}}\\ \sigma_{a_{d},a_{v,exp}} & \sigma_{a_{v,}exp}^{2} \end{array}\right]⊗(A\;or\;H)\right)$, where $\sigma_{a_{d}}^{2}$ is the DGE variance for HW in the mean model, $\sigma_{a_{v,}exp}^{2}$ is the DGE variance for HWU in the residual variance model, and $\sigma_{a_{d},a_{v,exp}}$ is the DGE covariance between HW and HWU; $Z_{s}$ is the design matrix representing IGEs in the mean model, while $a_{s}$ is used to denote the IGEs arising from social interactions among individuals within the same testing cage, and $a_{u}$ and $a_{s}$ are assumed to follow a multivariate normal distribution: $\left[\begin{array}{c} a_{u}\\ a_{s} \end{array}\right]\sim N\left(\left[\begin{array}{c} 0\\ 0 \end{array}\right],\left[\begin{array}{cc} \frac{1}{4}\sigma_{a_{d}}^{2} & \frac{1}{2}\sigma_{a_{d},a_{s}}\\ \frac{1}{2}\sigma_{a_{d},a_{s}} & \sigma_{a_{s}}^{2} \end{array}\right]⊗(A\;or\;H)\right)$, where $\sigma_{a_{s}}^{2}$ is the IGE variance, and $\sigma_{a_{d},a_{s}}$ is the covariance between DGEs and IGEs for HW; $Z_{t}$ is the design matrix for random cage effects (t) in the mean model (excluding cage effects in the H matrix): $t\sim N\left(0,I\sigma_{t}^{2}\right)$, $\sigma_{t\;}^{2}$ is the variance component for the random effects of the cages; $y\left(e\right)\;\mathrm{and}\;\Psi \left(e_{v}\right)$ are assumed to be independently normally distributed: $\left[\begin{array}{c} e\\ e_{v} \end{array}\right]\sim N\left(\left[\begin{array}{c} 0\\ 0 \end{array}\right],\left[\begin{array}{cc} W^{-1}\sigma_{\epsilon }^{2} & 0\\ 0 & W_{v}^{-1}\sigma_{\epsilon_{v}}^{2} \end{array}\right]\right)$, where $W=diag\left(\Psi^{-1}\right)\;\mathrm{and}\;{\;W}_{v}=diag\left(\frac{1-h_{i}}{2}\right)$, $\sigma_{\epsilon }^{2}\left(\sigma_{\epsilon_{v}}^{2}\right)$ is a scaled variance that was expected to be 1, the sire–dam DHGLM was fitted iteratively to update Ψ, diag$\left(W\right)$ and diag($W_{v}$) until the log-likelihood converged. The model fails to converge after incorporating the common environmental effects to a full-sib family group.

### 2.4. Genetic Parameters

#### 2.4.1. Estimation of Genetic Parameters for HW

The total breeding value (TBV), which incorporates both DGEs and IGEs, is calculated as follows [38]: $TBV=a_{d}+\left(n-1\right)a_{s}$, where $a_{d}={2a}_{u}$, and n−1 is the number of other individuals in the same cage (*n* = 36). The variance in TBV ($\sigma_{TBV}^{2}$) and phenotypic variance ($\sigma_{p}^{2}$) were calculated as shown in Table 1. In classical quantitative genetic theory, the heritability measure of the direct genetic variance relative to the phenotypic variance is calculated by $h^{2}=\frac{\sigma_{a_{d}}^{2}}{\sigma_{p}^{2}}$, where $\sigma_{a_{d}}^{2}=4\sigma_{a_{u}}^{2}$. By analogy, to express the heritable total variance to the phenotype variance, $T^{2}$ is introduced as the ratio of $\sigma_{TBV}^{2}$ to $\sigma_{p}^{2}$. In this study, $T^{2}$ only represents the heritable variance expressed on the scale of phenotypic variance among tested shrimp. A comparison of $T^{2}$ and $h^{2}$ gives a quick indication of the contribution of social effects to heritable variance. The correlation between DGEs and IGEs is given by $r_{{a_{d},a}_{s}}=\frac{\sigma_{{a_{d},a}_{s}}}{\sqrt{\sigma_{a_{d}}^{2}\sigma_{a_{s}}^{2}}}$.

#### 2.4.2. Estimation of Genetic Parameters for HWU

In the sire–dam DHGLM, the estimated genetic variance for sires was assumed to be equal to that of dams and set to one-quarter of the additive genetic variance. Hence, the additive genetic variance for HWU ($\sigma_{a_{v},exp}^{2}$) was equal to 4$\sigma_{a_{u_{v}},exp}^{2}$, where $\sigma_{a_{u_{v}},exp}^{2}$ represents the additive genetic variance contributed by the parents. Estimates for $\sigma_{a_{v},exp}^{2}$ for HWU were on the exponential scale (exp) and were converted to an additive scale ($\sigma_{a_{v}}^{2}$) using the extension of the equations of Mulder et al. [4], as derived by Sae-Lim et al. [13]. Heritability for HWU ($h_{v}^{2}$) on the additive scale was calculated as $h_{v}^{2}=\frac{\sigma_{a_{v}}^{2}}{2\sigma_{p}^{4}+3\sigma_{a_{v}}^{2}}$. The genetic coefficient of variation for HWU (${GCV}_{V}$) was calculated as ${GCV}_{V}=\frac{\sigma_{a_{v}}}{\sigma_{E}^{2}},where\;\sigma_{E}^{2}=\sigma_{e}^{2}-{2\sigma }_{a_{u}}^{2}.$ The correlation between HW and HWU is given by $r_{g}=\frac{\sigma_{a_{d},a_{v,exp}}}{\sqrt{\sigma_{a_{d}}^{2}\sigma_{a_{v,exp}}^{2}}}$. Approximated standard errors of $h_{v}^{2}$ and ${GCV}_{v}$ are not available in ASReml, and to our knowledge, have not been derived.

### 2.5. Cross Validation

Two genetic evaluation methods, pBLUP and ssGBLUP, were applied using genetic parameters estimated from A_NoIGE and H_NoIGE in ASReml. Due to convergence issues, the A_IGE and H_IGE models were not included in the cross-validation analysis. A 10-fold cross-validation approach with 20 replicates was conducted to evaluate prediction accuracy for both HW and HWU. Adjusted phenotypes for HW and HWU were calculated as $y_{i}^{*}={\hat{a}}_{d_{i}}+{\hat{t}}_{i}+{\hat{e}}_{i}$ and $\psi_{i}^{*}={\hat{a}}_{v_{i}}+{\hat{e}}_{v_{i}}$, using the solutions from A_NoIGE. Phenotypic data of genotyped samples were randomly divided into 10 groups. In each iteration, one group’s phenotypic data was withheld, and the remaining nine groups were used to calculate the estimated breeding values (EBV) or genomic EBV (GEBV). Prediction accuracy was evaluated using the correlation coefficient between adjusted phenotypes and EBV (GEBV). A correlation coefficient closer to 1 indicates higher prediction accuracy. Prediction bias was assessed by calculating the regression coefficient between phenotypic values and predicted GEBVs (or EBVs). A regression coefficient less than 1 indicates overestimation of GEBVs (or EBVs), whereas a value greater than 1 reflects underestimation.

## 3. Results

### 3.1. Descriptive Statistics

Descriptive statistics for the HW of *P. vannamei* are presented in Table 2. A total of 1440 individuals from 40 families were tested. After 55 days, 1335 individuals were harvested, resulting in an overall survival rate of 92.71%. The mean HW was 16.65 ± 2.64 g, with a range from 7.2 g (minimum) to 26.8 g (maximum). The coefficient of variation (CV) of HW for all individuals was 15.84%. The HW of female shrimp was slightly higher than that of males, with mean values of 17.10 g and 16.19 g, respectively. A boxplot of HW by family and cage is shown in Figure 2. The median HW for families and cages was 16.65 g and 16.51 g, respectively. The standard deviation (SD) and CV for family groups were slightly higher (SD = 1.73 g; CV = 10.40%) compared to cage groups (SD = 1.52 g; CV = 9.13%).

### 3.2. Genetic Parameters for HW and HWU

The variance components of HW and HWU are shown in Table 3. The additive genetic variance ($\sigma_{a_{d}}^{2}$) for HW was 0.139 (A_NoIGE), 0.119 (A_IGE), 0.139 (H_NoIGE), and 0.125 (H_IGE), respectively. In both the models with the A matrix and H matrix, incorporating IGEs reduced $\sigma_{a_{d}}^{2}$ by 10.07% to 14.39% compared to the no IGE models. Similarly, the heritability of HW decreased by 4.46% to 7.44% after incorporating IGEs, yet it remained moderately high, ranging from 0.622 to 0.673. $\sigma_{a_{d}}^{2}$ for HW based on the A matrix and H matrix were generally similar; however, in the IGE model, the estimate from the H matrix was 5.04% higher than that from the A matrix.

The heritability of HWU ($h_{v}^{2}$) ranged from 0.005 to 0.017, indicating low heritability. However, the coefficient of genetic variation for HWU (${GCV}_{v}$) ranged from 0.340 to 0.528, suggesting that using the residual variance in HW as a selection criterion to improve the HWU in *P. vannamei* is feasible. After accounting for IGEs, the $h_{v}^{2}$ of HWU increased by 150% to 240%, and the ${GCV}_{v}$ increased by 32.11% to 55.29%, indicating that including IGEs in the model could substantially enhance the accuracy and genetic gain in uniformity selection.

The correlation between HW and HWU varied significantly across different models. Without IGEs, the correlation ranged from −0.862 to −0.683, indicating a strong negative correlation between the mean and variance in HW. With IGEs, the correlation became a low positive range (0.196 to 0.275), with the lowest absolute genetic correlation observed in the H_IGE model at 0.196.

It is worth noting that the indirect genetic variance ($\sigma_{a_{s}}^{2}$) was 0.000136 (A) and 0.000187 (H), respectively. IGEs influenced other individuals within the cage (*n* = 36), resulting in an enhanced effect. Therefore, the actual variance produced by IGEs was 0.167 and 0.229 (${\left(n-1\right)}^{2}\sigma_{a_{s}}^{2}$), respectively, contributing 17.16% and 21.32% ($\frac{{\left(n-1\right)}^{2}\sigma_{a_{s}}^{2}}{\sigma_{TBV}^{2}}\times 100\%$) to $\sigma_{TBV}^{2}$. The H matrix could partition more IGEs than the A matrix. The variance in the cage effect ($\sigma_{t}^{2}$) for HW ranged from 0.008 ± 0.014 to 0.029 ± 0.012, accounting for 1.05% to 3.51% of the phenotypic variance. Incorporating IGEs reduced the variance component of the cage effect, indicating that the cage effect consists not only of environmental differences but also of interactions between individuals.

### 3.3. Prediction Accuracy

The predictive performance of the different models was evaluated, and the mean Pearson correlation coefficient between adjusted phenotypes and the predicted EBV (GEBV), along with bias, are presented in Table 4. The results show that the prediction accuracy of ssGBLUP (HW = 0.385; HWU = 0.084) was higher than that of pBLUP (HW = 0.362; HWU = 0.076) for both HW and HWU. Specifically, ssGBLUP improved prediction accuracy by 6.35% for HW and 10.53% for HWU compared to pBLUP. However, the bias for ssGBLUP (HW = 1.212; HWU = 1.425) were slightly higher than those for pBLUP (HW = 1.159; HWU = 1.253), suggesting a small trade-off in prediction bias despite the accuracy improvement.

## 4. Discussion

### 4.1. Genetic Parameters of HWU

In this study, the $h_{v}^{2}$ for HWU in *P. vannamei*, estimated using the sire–dam DHGLM model, ranged from 0.005 to 0.017, indicating low heritability. This result is consistent with uniformity estimates reported for livestock and aquatic animals, which typically range from 0 to 0.1 [39]. Sae-Lim et al. [13] estimated the genetic parameters for HWU in rainbow trout under two different environments, reporting heritability estimates for HWU ranging from 0.01 to 0.024. Similarly, García-Ballesteros et al. [6] found low heritability estimates (0.01 to 0.03) for HWU in *P. vannamei* across nucleus and commercial populations. The low heritability estimates for HWU are primarily due to the inability to directly measure individual dispersion, which must be inferred through residuals. Residuals have a mean close to zero and are symmetrically distributed, providing limited information from individual data. However, analyzing multiple families allows the significant differences between families to be quantified using statistical methods, enabling more effective variance estimation [9]. In this context, increasing repeated records at the individual level or expanding family size can enhance the model’s estimation capability, thereby achieving higher heritability estimates [8,39,40]. This facilitates the acquisition of more accurate breeding values, leading to a faster genetic gain in uniformity traits.

Additionally, the genetic coefficient of variation for HWU (${GCV}_{v}$) was 0.340 to 0.528, indicating that using the residual variance in HW as a selection criterion to improve the HWU in *P. vannamei* is feasible. The results of this study are similar to previous reports. For example, Marjanovic et al. [7] conducted a genetic evaluation of the uniformity of growth traits in Nile tilapia, reporting a genetic coefficient of variation ranging from 0.42 to 0.58. Similarly, Garcia et al. [2] reported moderate coefficients of variation (${GCV}_{v}$: 0.17 to 0.35) for HW and HWU in *P. vannamei* across two different environments. In summary, these results indicate that genetic selection for improving uniformity has promising prospects.

### 4.2. The Influence of IGEs

The inclusion of IGEs in the model revealed their significant contribution to the total genetic variance for HW in *P. vannamei*, accounting for 17.16% (A matrix) and 21.32% (H matrix). This highlights the significant influence of social interactions on growth traits, consistent with findings in other species such as Nile tilapia and Atlantic cod [23,41]. Previous studies report that IGEs can explain 6% to 98% of the total genetic variance, depending on the species, traits, and environmental conditions [42,43].

In this study, incorporating IGEs reduced the direct genetic variance while increasing the genetic variance for HWU. This redistribution highlights that part of the variation previously attributed to direct effects or residual variance can be better captured by IGEs, improving the precision of genetic evaluations. Consequently, heritability estimates for HWU and its ${GCV}_{v}$ showed substantial increases, further demonstrating the utility of IGEs in modeling uniformity traits. The accurate separation of environmental factors from genetic effects remains a challenge when estimating residual variance [44]. Without IGEs, its contribution may be confounded with environmental effects, leading to biased estimates. Explicitly accounting for IGEs reduces this bias and provides a clearer partitioning of genetic and environmental variances. To further evaluate the significance of IGEs, an investigation was performed on the IGE model. Specifically, a 10-fold cross-validation with five replicates was used to compare the impact of IGEs on the prediction accuracy of HW and HWU (A_IGE vs. A_NoIGE). The results showed that for HW, the prediction accuracy increased from 0.313 (A_NoIGE) to 0.371 (A_IGE), an improvement of 18.53%. For HWU, the prediction accuracy increased from 0.038 to 0.046, representing an increase of 21.05%. These findings highlight the importance of IGEs in accurately capturing social interaction effects and redistributing genetic variance, thereby improving genetic evaluation precision.

In the breeding programs of *P. vannamei*, the 3FAM design is widely used for estimating IGEs [43]. However, its strict family control requirements increase experimental complexity, particularly in individual identification and tracking, leading to high management costs and limiting its application in large-scale breeding. For shrimp breeding programs, the integration of IGEs is particularly feasible and advantageous. Compared to livestock and fish species [22,28,42], *P. vannamei* has larger full-sib family sizes, shorter production cycles, and smaller body sizes, all of which facilitate IGE estimation [43]. These unique biological and production traits make IGE-based selection a promising approach for improving HWU and achieving greater genetic gains.

### 4.3. Genetic Correlation Between HW and HWU

In studies on uniformity in livestock and aquaculture, log or Box–Cox transformations are often applied to phenotypic data to eliminate potential mean–variance relationships and reduce model bias caused by scale effects [13,16,39]. However, no scale effect was observed in HW in this study, so untransformed data were used for analysis. Without incorporating IGEs, their contributions were confounded with DGEs, resulting in an overestimation of DGEs and an abnormally high proportion of total genetic variance [7,28]. In such cases, the genetic correlation between HW and HWU was strongly negative (A: −0.862; H: −0.683), likely reflecting overestimated DGEs rather than the true genetic relationship.

Incorporating IGEs into the model improved the partitioning of DGEs and IGEs. This adjustment reallocated previously overestimated DGEs to IGEs, which lowered DGE estimates without significantly changing residual variance. Consequently, the genetic correlation between HW and HWU weakened (A: 0.203; H: 0.117), consistent with prior findings in *P. vannamei* (0.17 to 0.34) [2,6]. This shift underscores the influence of the positive correlation between DGEs and IGEs (0.509 ± 0.264 to 0.578 ± 0.328) on the genetic relationship between HW and HWU. When DGEs and IGEs are positively correlated, individuals with higher DGEs (e.g., greater potential for growth) also exhibit stronger IGEs, thereby promoting growth among their groupmates [43]. This dynamic reduces in HWU as the group grows more uniformly due to indirect effects. Incorporating IGEs ensures that cooperative behaviors are properly attributed, reducing the bias in DGE estimates and highlighting IGEs’ positive contribution to uniformity (HWU). These results emphasize the importance of including IGEs in genetic evaluation models to accurately assess the genetic relationship between HW and HWU. Precise partitioning of genetic effects clarifies the interplay between traits and enhances the model’s capacity to evaluate individual interactions and genetic effect distributions. This provides a more biologically realistic understanding of how genetic and group-level dynamics influence growth and uniformity traits.

### 4.4. The Prediction Accuracy

This study showed that ssGBLUP outperformed pBLUP, improving by 10.53% for HWU and 6.35% for HW. Similarly, in selective breeding for body weight in Chinese shrimp (*Fenneropenaeus chinensis*), the prediction accuracy of ssGBLUP under biosecurity restrictions improved by 29.60% over the A matrix, further validating the advantages of genomic information incorporated in the H matrix [45]. In studies on uniformity-related traits, Sae-Lim et al. [16] demonstrated that genetic evaluations using the H matrix outperformed those using the A matrix, with improvements ranging from 41.1% to 78.1%, and showed moderate improvements of 1.3% to 13.9% after log-transforming body weight to adjust for scale effects. ssGBLUP integrates genotype data into the estimation of breeding values, utilizing genomic information through identity by state (IBS) to more precisely characterize the genetic relationships between individuals, and allowing for a more accurate estimation of genetic variance and genetic covariance, thereby enhancing the accuracy of genetic effect estimation [46].

In this study, although 84.72% (*n* = 1220) of individuals were genotyped, data structure limitations and reliance on sire–dam DHGLM models limited the full use of genomic information, hindering further improvements in prediction accuracy. Similar limitations were also reported by Mulder et al. [47] and Sell-Kubiak et al. [48], where genetic parameter estimates for mean and residual variance were comparable when using both the pedigree-based relationship matrix (PRM) and the genomic relationship matrix (GRM). Sell-Kubiak et al. [48] further emphasized that the H matrix outperforms the A matrix only with a sufficiently large reference population, recommending at least 2000 genotyped individuals. These findings suggest that while the H matrix is promising for low-heritability traits like HWU, its effectiveness depends on a large-scale reference population and comprehensive genomic data, posing challenges in practical breeding applications.

Due to convergence issues, the A_IGE and H_IGE models were not included in the cross-validation analysis, making it difficult to compare the impact of genomic information and IGE integration on uniformity prediction accuracy. However, relevant studies show that genomic information can effectively improve the prediction accuracy of total genetic effect (TGE), which includes both DGEs and IGEs. Poulsen et al. [49] and Leite et al. [50] found that incorporating genomic information improved TGE prediction accuracy by 5.9% and 12% for pigs’ growth rate and sows’ skin lesion traits, respectively. Therefore, it is hypothesized that integrating genomic information and IGEs may enhance the accuracy of breeding value estimation for HWU and provide deeper insights into its genetic mechanisms. 

## 5. Conclusions

This study integrated genomic information and IGEs to estimate genetic parameters for HW and HWU in *P. vannamei*, providing a theoretical foundation for future research. HWU exhibited low heritability but moderate genetic variation, indicating its potential for selective breeding. Incorporating IGEs into genetic parameter estimation improved the partitioning of genetic and environmental effects, leading to more accurate evaluations of genetic parameters for HWU. Given the large full-sib family sizes, small body size, and short production cycles in *P. vannamei*, accounting for IGEs has a great potential in shrimp breeding programs. Additionally, integrating genomic information using ssGBLUP enhanced prediction accuracy for HWU compared to traditional methods.

## Figures and Tables

**Figure 1 biology-14-00328-f001:**
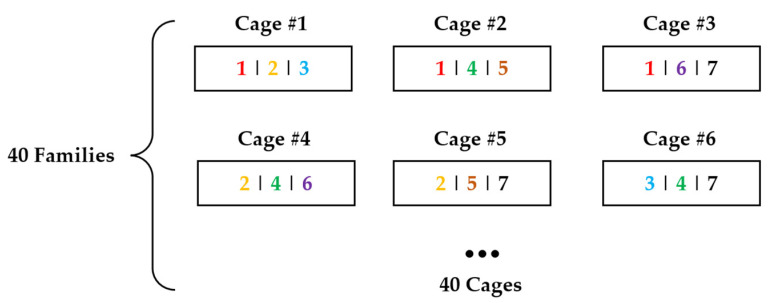
Graphical representation of the design with three families per group (3FAM). Each block represents a net cage. The numbers in each block indicate the family composition of each cage. ‘1|2|3’ represents families 1, 2, and 3.

**Figure 2 biology-14-00328-f002:**
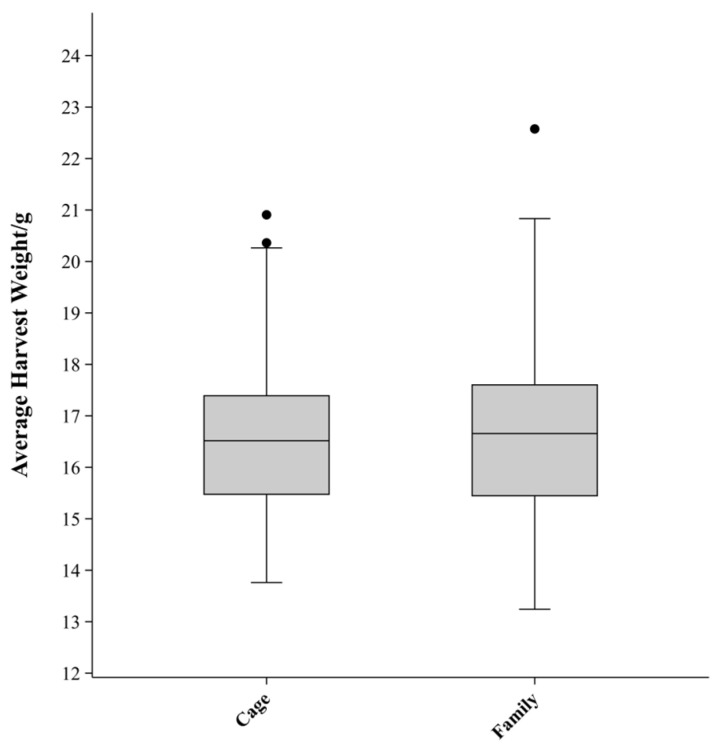
Boxplot of harvest weight by family and cage of *P. vannamei*.

**Table 1 biology-14-00328-t001:** Genetic parameters and variance components for harvest weight.

Variance Components	Formula	Explanation
$\mathrm{Variance}\;\mathrm{in}\;\mathrm{TBV}\;(\sigma_{TBV}^{2}$)	$\sigma_{TBV}^{2}=\sigma_{a_{d}}^{2}+2\left(n-1\right)\sigma_{{a_{d},a}_{s}}+{\left(n-1\right)}^{2}\sigma_{a_{s}}^{2}$	
$\mathrm{Phenotypic}\;\mathrm{variance}\;(\sigma_{p}^{2}$)	$\sigma_{p}^{2}=\sigma_{a_{d}}^{2}+\sigma_{t\;}^{2}+\sigma_{e\;}^{2}$	A_NoIGE and H_NoIGE
	$\sigma_{p}^{2}=\sigma_{a_{d}}^{2}+\left(n-1\right)\sigma_{a_{s}}^{2}+\sigma_{t}^{2}+\sigma_{e}^{2}$	$A\text{\_}\mathrm{IGE},\;H\text{\_}\mathrm{IGE}\;(\mathrm{without}\;\sigma_{t}^{2}$)

**Table 2 biology-14-00328-t002:** Descriptive statistics of harvest weight of *P. vannamei*.

Sex	N1	N2	Mean of M1BW/g	Mean/g	Minimum/g	Maximum/g	Standard Deviation	Coefficient of Variation/%	Survival Rate/%
All	1440	1335	4.81	16.65	7.2	26.8	2.64	15.84	92.71
Male	/	665	4.77	16.19	7.2	26.1	2.47	15.27	/
Female	/	670	4.88	17.10	8.1	26.8	2.72	15.91	/

Number of individuals at stocking—N1; number of individuals at harvest—N2; stocking body weight at tagging—M1BW.

**Table 3 biology-14-00328-t003:** Genetic parameters for harvest weight and its uniformity in *P. vannamei* based on the A matrix and H matrix.

Parameters	A Matrix		H Matrix	
	A_NoIGE	A_IGE	H_NoIGE	H_IGE
$\sigma_{a_{d}}^{2}$	0.139 ± 0.033	0.119 ± 0.031	0.139 ± 0.034	0.125 ± 0.033
$\sigma_{a_{ds}}$	_	0.00233 ± 0.0012	_	0.00246±0.00133
$\sigma_{a_{s}}^{2}$	_	0.000136 ± 0.000117	_	0.000187 ± 0.000088
$\sigma_{a_{v}}^{2}$	0.009	0.022	0.008	0.019
$\sigma_{a_{v},exp}^{2}$	0.109	0.246	0.137	0.228
$\sigma_{e}^{2}$	0.520	0.517	0.520	0.521
$\sigma_{e,exp}^{2}$	0.229	0.246	0.226	0.242
$\sigma_{t}^{2}$	0.029 ± 0.012	0.008 ± 0.014	0.028 ± 0.011	-
$\sigma_{p}^{2}$	0.826	0.768	0.826	0.782
$\sigma_{TBV}^{2}$	_	0.971	_	1.072
$h^{2}$	0.672	0.622	0.673	0.643
$T^{2}$	_	1.264	_	1.380
$h_{v}^{2}$	0.005	0.017	0.006	0.015
${GCV}_{v}$	0.340	0.528	0.383	0.506
$r_{a_{ds}}$	_	0.578 ± 0.328	_	0.509 ± 0.264
$r_{g}$	−0.862 ± 0.627	0.203 ± 0.213	−0.683 ± 0.427	0.117 ± 0.230

The direct genetic variance in harvest weight—$\sigma_{a_{d}}^{2}$; the direct–indirect genetic covariance in harvest weight—$\sigma_{a_{ds}}$; the indirect genetic variance in harvest weight—$\sigma_{a_{s}}^{2}$; additive genetic variance for uniformity—$\sigma_{a_{v}}^{2}$; additive genetic variance for uniformity on the exponential scale (exp)—$\sigma_{a_{v},exp}^{2}$; residual variance—$\sigma_{e}^{2}$; residual variance on the exponential scale (exp)—$\sigma_{e,exp}^{2}$; variance in random effects of cages—$\sigma_{t}^{2}$; phenotypic variance in harvest weight—$\sigma_{p}^{2}$; total breeding value variance in harvest weight—$\sigma_{TBV}^{2}$; heritability of harvest weight—$h^{2};$ heritability of uniformity—$h_{v}^{2}$; genetic coefficient of variation for uniformity—${GCV}_{v}$; genetic correlation between DGEs and IGEs for harvest weight—$r_{a_{ds}}$; genetic correlation between harvest weight and its uniformity—$r_{g}$.

**Table 4 biology-14-00328-t004:** Prediction accuracy and bias of harvest weight and its uniformity in *P. vannamei* using different models.

	Harvest Weight		Uniformity of Harvest Weight	
	Accuracy	Bias	Accuracy	Bias
pBLUP	0.362	1.159	0.076	1.253
ssGBLUP	0.385	1.212	0.084	1.425

## Data Availability

The data that support the findings of this study are available from the corresponding author upon reasonable request.

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
