# Peer review of "Genomic Evaluation of Harvest Weight Uniformity in Penaeus vannamei Under a 3FAM Design Incorporating Indirect Genetic Effect"

_biology, 2025, doi:10.3390/biology14040328_

Round 1

Reviewer 1 Report (Previous Reviewer 1)

Comments and Suggestions for Authors

I agree that "this study is still in the exploratory phase and has a long way to go before it can be applied in practice." I suggest that they include this statement in the manuscript.

Author Response

Comments 1: I agree that "this study is still in the exploratory phase and has a long way to go before it can be applied in practice." I suggest that they include this statement in the manuscript.

Response 1: Thank you for your valuable feedback. I have added the statement, "This study is still in the exploratory phase and has a long way to go before it can be applied in practice," to the conclusion section of the manuscript. (L489-490)

Reviewer 2 Report (New Reviewer)

Comments and Suggestions for Authors

The paper is well structured; however, I suggest adding some details to the methodology, as it should be more oriented towards the field of aquaculture in order to develop selection programs.

I suggest changing to Penaeus vannamei.

Define FAM3 in the text.

Did the 40 families have a pre-selected phenotypic characteristic?

Line 158: Please specify size and/or weight.

Line 159: Visible Implant Elastomer (VIE) markers (Northwest Marine Technology, NMT Inc, Shaw Island, WA) – Where was the tagging performed?

Lines 161-163: Improve the clarity of the methodology in the division of families.

Line 172: It should be m³.

Line 183: Change to mL.

Line 192: This analysis could be included as supplementary material.

Line 195: Indicate the version of R used.

Was the distribution of the cages within the pond random?

In the family division, was the sex ratio evaluated? The females showed greater growth, which could be an effect of selection or a natural occurrence.

Could you specify some characteristics of the water quality parameters in the system (e.g., temperature, oxygen, pH, etc.)?

Was there any recirculation system in place?

How was feeding conducted, and what was the food provided, and how often?

Was DNA quality and concentration evaluated? For example, was it assessed using an agarose gel, and was concentration measured using a Thermo Scientific NanoDrop 2000?

How was social interaction assessed?

Could you include this as an error?

Results

Table 2: The "number of families" column could be removed as it is common for all.

Were there differences between families and cages?

Was the effect of sex left undiscussed? Does it have any implications for selection and the effect of interaction?

They mention environmental effects, but which ones are they referring to if they do not mention the growing conditions?

Author Response

Comments 1: I suggest changing to Penaeus vannamei.

Response 1: Thank you for your suggestion. I have revised all instances of 'Litopenaeus vannamei' in the manuscript to 'Penaeus vannamei'.

Comments 2: Define FAM3 in the text.

Response 2: Thank you for your feedback. 3FAM is defined as a grouping design with three families per group, and this has been updated in the manuscript. (L164, Figure 1)

Comments 3: Did the 40 families have a pre-selected phenotypic characteristic?

Response 3: No. After each family reaches the P60 size specification, 36 individuals are randomly selected from each family for the 3FAM experimental design.

Comments 4: Line 158: Please specify size and/or weight.

Response 4: Thank you for your feedback. The P60 size specification of approximately 4.8g, and this has been updated in the manuscript. (L160)

Comments 5: Line 159: Visible Implant Elastomer (VIE) markers (Northwest Marine Technology, NMT Inc, Shaw Island, WA) – Where was the tagging performed?

Response 5: Thanks. The sixth abdominal segment of each shrimp was tagged using Visible Implant Elastomer (VIE) markers (Godin et al., 1996). (L160-162)

Reference:

Godin, D. M., Carr, W. H., Hagino, G., Segura, F., Sweeney, J. N., & Blankenship, L. (1996).  Evaluation of a fluorescent elastomer internal tag in juvenile and adult shrimp Penaeus vannamei. Aquaculture, 139(3-4), 243-248.

Comments 6: Lines 161-163: Improve the clarity of the methodology in the division of families.

Response 6: Thank you for your feedback. To more clearly illustrate the family division, we have supplemented Figure 1 to explain the design of the 3FAM experiment. (L164)

Comments 7: Line 172: It should be m³. Line 183: Change to mL.

Response 7: Thank you for your suggestion. The revision has been made in the manuscript. (L176, 197)

Comments 8: Line 192: This analysis could be included as supplementary material.

Response 8: As some studies are still unpublished, we are unable to make the full dataset publicly available currently. For this, we sincerely apologize. We are committed to providing relevant data upon reasonable request to support the review process or further research. Additionally, once the related studies are published, we will ensure that the data is made accessible to facilitate academic exchange.

Comments 9: Line 195: Indicate the version of R used.

Response 9: Thanks. The modifications have been made as per your request. (L209)

Comments 10: Was the distribution of the cages within the pond random?

Response 10: Yes. Forty cylindrical cages were positioned in the pond, with 20 cages placed along each side of a central cement walkway.

Comments 11: In the family division, was the sex ratio evaluated? The females showed greater growth, which could be an effect of selection or a natural occurrence.

Response 11: Thanks. There is sexual dimorphism in shrimp, which is a natural phenomenon. Research has reported that sexual dimorphism in cultured Penaeus vannamei starts to appear when shrimp reach 10-17g (Chow et al., 1991; Pérez-Rostro et al., 1999). These size differences are most likely caused by different biological requirements at the onset of sexual maturation (Pérez-Rostro et al., 2003). At the beginning of the experiment, the small size of the shrimp makes it challenging to accurately distinguish between female and male individuals. Given that the average weight of female shrimp is greater than that of males, we incorporated the sex factor as fixed effect on the genetic assessment in the DHGLM model to decrease the influence of sex effect.

Reference:

Chow, S., Sandifer, P.A., 1991. Differences in growth, morphometric traits, and male sexual maturity among Pacific white shrimp, Penaeus vannamei, from different commercial hatcheries. Aquaculture 92, 165–178.

Pérez-Rostro, C.I., Ramirez, J.L., Ibarra, A.M., 1999. Maternal and cage effects on genetic parameter estimation for Pacific white shrimp Penaeus vannamei Boone. Aquac. Res. 30 (9), 681–693.

Pérez-Rostro, C.I., Ibarra, A.M., 2003. Heritabilities and genetic correlations of size traits at harvest size in sexually dimorphic Pacific white shrimp (Litopenaeus vannamei) grown in two environments. Aquac. Res. 34 (12), 1079–1085.

Comments 12: Could you specify some characteristics of the water quality parameters in the system (e.g., temperature, oxygen, pH, etc.)?

Response 12: The water quality parameters in the system are as follows: pH range from 8.04 to 8.94, salinity ranges from 26 to 31‰. Water temperature during the day ranges from 21.8°C to 34°C, while at night it ranges from 21.4°C to 28.2°C. Dissolved oxygen concentrations range from 5.2 to 8.7 mg/L, and total alkalinity ranges from 118.08 to 200.72 mg/L. Ammonia nitrogen concentrations range from 0.01 to 0.27 mg/L, and nitrite concentrations range from 0.01 to 1.22 mg/L. (L178-183)

Comments 13: Was there any recirculation system in place?

Response 13: Thanks. This experiment did not employ a recirculating water system. Due to the earthen pond aquaculture model, evaporation leads to a daily decline in water level of 15%-20%. Water is added each night to restore the water level to the normal line to ensure an optimal culture environment.

Comments 14: How was feeding conducted, and what was the food provided, and how often?

Response 14: Thanks. During the juvenile stage, each cage received 0.5 g of shrimp flakes per meal. As the shrimp grew, the feed was progressively transitioned to stage-specific Charoen Pokphand Foods (CPF) Pacific white shrimp compound feed. Feeding was conducted five times daily at 7:00, 10:00, 14:00, 17:00, and 20:30, with feed mixing required before each feeding. The feeding amount was assessed based on the shrimp's daily consumption and adjusted accordingly to ensure optimal feeding management.

Comments 15: Was DNA quality and concentration evaluated? For example, was it assessed using an agarose gel, and was concentration measured using a Thermo Scientific NanoDrop 2000?

Response 15: Genomic DNA was extracted from muscle using a DNeasy 96 Blood & Tissue Kit (Qiagen, Shanghai, China). The integrity of DNA was detected by agarose gel electrophoresis. Nanodrop2000 (Thermo Scientific, Wilmington, DE, USA) was used to detect the concentrations of extracted DNA.

Comments 16: How was social interaction assessed?

Response 16: Thank you for your feedback. Social interactions between individuals can have a genetic component, known as indirect genetic effects (IGE), where an individual's genotype influences the phenotypes of its social partners. To estimate IGE, experimental designs require individuals to be distributed across multiple groups. This is because the data structure derived from traditional communal rearing environments lacks genetic group segregation, making it inadequate to disentangle DGE from IGE. Advanced methodologies propose using groups composed of two or three families combined with extended animal models that account for kinship coefficients. For example, the variance of indirect genetic effects (IGE) on harvest weight in Pacific white shrimp was estimated using an experimental design, which comprised 105 small groups, with each group containing 15 shrimp (Luan et al., 2015). In Nile tilapia, an optimal design of multiple blocks of 11 families was used to estimate IGE on harvest weight (Khaw et al., 2016; Bijma, 2010).

Reference:

Luan, S.; Luo, K.; Chai, Z.; Cao, B.; Meng, X.; Lu, X.; Liu, N.; Xu, S.; Kong, J. An analysis of indirect genetic effects on adult body weight of the Pacific white shrimp Litopenaeus vannamei at low rearing density. Genet Sel Evol. 2015, 47, 1-8.

Khaw HL, Ponzoni RW, Yee HY, Aziz MAB, Bijma P. Genetic and non-genetic indirect effects for harvest weight in the GIFT strain of Nile tilapia (Oreochromis niloticus). Aquaculture. 2016;450:154–61.

Bijma P. Estimating indirect genetic effects: precision of estimates and optimum designs. Genetics. 2010;186:1013–28.

Comments 17: Could you include this as an error?

Response 17: Thank you for your feedback. In the context of the classical quantitative genetic model where the trait value of an individual is decomposed into the heritable effect of its genotype and a residual labelled as environment: P=G+E, the E-term is partly heritable when IGEs occur (Bijma, 2014). Therefore, when IGEs are not accounted for in the model, their effects may be classified as unestimable environmental factors and included in the residual term. In studies on livestock and aquaculture species, residual variance is often used as an indicator of uniformity (Iung et al., 2020). If the influence of IGEs is ignored, it may lead to bias in the estimation of residual variance, thereby affecting the accurate estimation of genetic parameters for uniformity.

Reference:

Bijma, P. (2014). The quantitative genetics of indirect genetic effects: a selective review of modelling issues.Heredity,112(1), 61-69.

Iung, L.H.D.S.; Carvalheiro, R.; Neves, H.H.D.R.; Mulder, H.A. Genetics and genomics of uniformity and resilience in livestock and aquaculture species: a review. J Anim Breed Genet. 2020, 137, 263-280.

Comments 18: Results. Table 2: The "number of families" column could be removed as it is common for all.

Response 18: Thank you for your suggestion. The revision has been made in the manuscript. (Table 2)

Comments 19: Were there differences between families and cages?

Response 19: Thank you for your feedback. We do not fully understand your specific meaning, but we will attempt to respond from two perspectives. On one hand, the distinction between family lines and cages in experiment design. Each cage contains three families (with any two families combined appearing in only one cage), and all cages are placed within the same farming pond, ensuring that all individuals grow under identical environmental conditions. Comparing the traditional communal rearing, the 3FAM grouping experimental design can effectively distinguish between DGE and IGE on growth (Poulsen et al., 2020; Nielsen et al., 2014; Khaw et al., 2016; Luan et al., 2015).

On the other hand, from the perspective of the interaction between families and cages. The three replicate of the same family exhibited differences IGE across different cages. However, due to the large number of families and cages, and they were not fully crossed, their interaction could not be included as a fixed effect to eliminate its influence. In this study, family and cage were treated as random effects to account for their influence and to minimize reduce confounding.

Reference:

Poulsen, B. G.; Ask, B.; Nielsen H. M.; Ostersen, T.; Christensen, O. F. Prediction of genetic merit for growth rate in pigs using animal models with indirect genetic effects and genomic information. Genet Sel Evol. 2020, 52, 58.

Nielsen HM, Monsen BB, Odegård J, Bijma P, Damsgård B, Toften H, et al. Direct and social genetic parameters for growth and fin damage traits in Atlantic cod (Gadus morhua). Genet Sel Evol. 2014;46:5.

Khaw HL, Ponzoni RW, Yee HY, Aziz MAB, Bijma P. Genetic and non-genetic indirect effects for harvest weight in the GIFT strain of Nile tilapia (Oreochromis niloticus). Aquaculture. 2016;450:154–61.

Luan, S.; Luo, K.; Chai, Z.; Cao, B.; Meng, X.; Lu, X.; Liu, N.; Xu, S.; Kong, J. An analysis of indirect genetic effects on adult body weight of the Pacific white shrimp Litopenaeus vannamei at low rearing density. Genet Sel Evol. 2015, 47, 1-8.

Comments 20: Was the effect of sex left undiscussed? Does it have any implications for selection and the effect of interaction?

Response 20: Thank you for your feedback. In Penaeus vannamei, female shrimp are generally larger than males, but the weight difference is relatively small, with an average difference of less than 1 g in this study. Compared to other shrimp species like Macrobrachium rosenbergii, P. vannamei does not exhibit a pronounced social hierarchy between males and females. Instead, competitive behavior is likely driven primarily by size differences. Consequently, intense competition between individuals of different sizes may lead to a significant increase in the coefficient of variation of body weight at the cage level. Using the current data, we performed further statistical analysis and determined that the correlation coefficient between the sex ratio and the coefficient of variation in body weight, both calculated at the cage level and analyzed across 40 cages, was only 0.12. This indicates a very weak correlation, suggesting that sex had no significant impact on interaction and overall weight in this experiment. Furthermore, our findings support the idea that shrimp exhibit a cooperative tendency, as evidenced by a positive correlation between DGE and IGE (0.578 ± 0.328). Based on this, we infer that there is no significant social interaction between males and females in P. vannamei.

Reference:

Alemu, S. W., Berg, P., Janss, L., & Bijma, P. (2016). Estimation of indirect genetic effects in group‐housed mink (Neovison vison) should account for systematic interactions either due to kin or sex. Journal of Animal Breeding and Genetics133(1), 43-50.

Comments 21: They mention environmental effects, but which ones are they referring to if they do not mention the growing conditions?

Response 21: This study classifies environmental influences into three categories: macro-environmental factors (e.g., ecosystem-scale gradients like latitudinal temperature zones and marine-freshwater salinity contrasts), micro-environmental factors (e.g., microhabitat-specific fluctuations including physiological oscillations and unpredictable microscale variations), and social environmental factors (e.g., interspecific interactions such as cooperative behaviors and competitive dynamics). In statistical models, we typically include fixed effects for the systemic environment and common environmental effects to account for known environmental influences. However, certain environmental factors (such as social interactions and microenvironmental fluctuations) may be difficult to measure directly or estimate accurately, and thus are included in the residual component, affecting the model's ability to parse genetic effects. This study introduces indirect genetic effects (IGE) into the model to more accurately estimate the genetic effects on phenotypic variation (homogeneity) caused by differences in individuals' sensitivity to their microenvironments within the residuals.

This manuscript is a resubmission of an earlier submission. The following is a list of the peer review reports and author responses from that submission.

Round 1

Reviewer 1 Report

Comments and Suggestions for Authors

The paper examines the integration of genomic information and indirect genetic effects (IGE) to estimate genetic parameters for uniformity of harvest weight (HWU) in Pacific white shrimp (Litopenaeus vannamei). Using a 3FAM experimental design and double hierarchical generalized linear models (DHGLM), the study explores genetic variations in HWU, revealing its low heritability but substantial genetic coefficient of variation, suggesting potential for selective breeding improvements.

In general, this is a well conducted study. My major concern is the low heritability of HWU. Although the incoporation of IGE into the model increases heritability by 150% - 240%, it only raises heritability from 0.005 to 0.017 or from 0.006 to 0.015. In addition, while the genetic coefficient of variation (GCV) ranged from 0.340 to 0.528, it may not suggest that using the residual variance of HW as a selection criterion for improving the HWU is feasible. Because the numerator of GCV, σ2av , is so small, small changes in denominator will have a great impact on GCV. Consequently, I am not convinced that incorporating IGE and utilizing genomic selection methods can enhance genetic evaluation accuracy for HWU. Perhaps, the study demonstrates that the effective management during shrimp culture is more important for increasing HWU.

Author Response

Comments: In general, this is a well conducted study. My major concern is the low heritability of HWU. Although the incorporation of IGE into the model increases heritability by 150% - 240%, it only raises heritability from 0.005 to 0.017 or from 0.006 to 0.015. In addition, while the genetic coefficient of variation (GCV) ranged from 0.340 to 0.528, it may not suggest that using the residual variance of HW as a selection criterion for improving the HWU is feasible. Because the numerator of GCV, σ2av, is so small, small changes in denominator will have a great impact on GCV. Consequently, I am not convinced that incorporating IGE and utilizing genomic selection methods can enhance genetic evaluation accuracy for HWU. Perhaps, the study demonstrates that the effective management during shrimp culture is more important for increasing HWU.

Response: Thanks for your thoughtful comments. Indeed, we acknowledge that this study is still in exploratory phase, and there is still a long way to go before it can be applied in practice.

Currently, residual variance and the DHGLM model remain the main parameters and models for studying animal uniformity (Sae-Lim et al., 2015, Garcia et al., 2021, García-Ballesteros et al., 2021). Previous genetic evaluations of uniformity in livestock and aquatic animals have shown that the estimated heritability () for uniformity typically ranged from 0 to 0.1, while the estimated genetic coefficient of variation () ranged from 0 to 0.86 (Iung et al., 2020). Our results are consistent with these studies. However, the traditional DHGLM model struggles to accurately assess uniformity when the number of families and individuals is small. To address this, our study built upon previous work by using a 3FAM breeding design and incorporating indirect genetic effects (IGE) and the H matrix into the model. This improvement allowed us to effectively partition the indirect genetic effects in the environment, significantly increasing the heritability of uniformity, which indicates that this breeding model enhances the accuracy of residual predictions. In future research, we plan to further expand the family number and utilize repeated records data, with the aim of further improving the estimated heritability and the prediction accuracy of uniformity.

Additionally, as you mentioned, this study suggests that effective management in shrimp farming may be more important than genetic factors in improving HWU. Common practices in aquaculture, such as size grading, aim to reduce size differences and ensure uniformity within the population, but they also increase management complexity and costs. This study conducted a genetic evaluation of uniformity, and the results suggest that uniformity traits have certain breeding potential. By selecting for uniformity traits, it may help improve genetic progress in shrimp farming, potentially increasing production efficiency and possibly reducing management costs.

Reference:

García-Ballesteros, S.; Villanueva, B.; Fernández, J.; Gutiérrez, J.P.; Cervantes, I. Genetic parameters for uniformity of harvest weight in Pacific white shrimp (Litopenaeus vannamei). Genet Sel Evol. 2021, 53, 1-9.

Garcia, B.F.; Montaldo, H.H.; Iung, L.H.; Carvalheiro, R. Effect of harvest weight and its uniformity on survival in Litopenaeus vannamei reared in different systems. Aquaculture 2021, 531, 735891.

Sae-Lim, P.; Kause, A., Janhunen, M.; Vehviläinen, H.; Koskinen, H.; Gjerde, B.; Lillehammer, M.; Mulder, H.A. Genetic (co) variance of rainbow trout (Oncorhynchus mykiss) body weight and its uniformity across production environments. Genet Sel Evol. 201547, 1-10. 

Iung, L.H.D.S.; Carvalheiro, R.; Neves, H.H.D.R.; Mulder, H.A. Genetics and genomics of uniformity and resilience in livestock and aquaculture species: a review. J Anim Breed Genet. 2020137, 263-280. 

Reviewer 2 Report

Comments and Suggestions for Authors

Reviewer Report

Manuscript id: biology-3413836

Manuscript entitle: “Genetic Analysis of Harvest Weight Uniformity in Litopenaeus vannamei Incorporating Indirect Genetic Effects through Ge-nomic Information”

 In the manuscript author reports Pacific white shrimp (Litopenaeus vannamei) is a crucial species in global aquaculture, but significant variation in harvest weight (HW) within families im-pacts farm productivity and profitability. Improving the uniformity of harvest weight (HWU) is essential for optimizing production efficiency and economic returns. Social interactions in both natural and farm environments significantly affect traits like growth, feed intake, and mortality, causing performance variability. This study employed a 3FAM design and DHGLM model, integrating genomic information and indirect genetic effects (IGE), to estimate the genetic parameters for HWU. Our findings highlight the importance and feasibility of integrating indirect genetic effects into genetic assessments of HWU in shrimp. Furthermore, the inclusion of genomic data enhances the prediction accuracy of the model. These results will provide a theoretical foundation and offer valuable gu the pacific idance for the breeding of new shrimp varieties.

Major revision is required for improvement

In the abstract section

In the abstract section there are not need to mention abbreviation of short for the should be start from introduction

In the material method section

Sample and experimental design

While the study highlights the advantage of using the 3FAM design for indirect genetic effects (IGE), it does not provide sufficient detail on how the cages were assigned or randomized. The lack of randomization procedures may introduce bias.

The trial duration (55 days) seems short, given the objectives to evaluate growth uniformity, which may benefit from longer observation periods to capture full genetic and environmental interactions.

In the Genotype and quality control

The study does not address potential limitations of using the 55K SNP Panel or whether higher-density panels would improve accuracy for such low-heritability traits.

In the statistical analysis

The study faced convergence issues in the models incorporating IGE (A_IGE and H_IGE), which were subsequently excluded from cross-validation. This limitation weakens the strength of conclusions drawn about IGE's influence.

The mathematical equations for variance components are provided but could benefit from a clearer explanation or visual representation for accessibility to a broader audience.

In the result section

Genetic Parameter Estimation

The heritability estimates for HWU (0.005–0.017) are exceptionally low, and while the study acknowledges this, it does not sufficiently discuss potential reasons, such as environmental noise or measurement errors.

The large increase in HWU heritability (150–240%) after incorporating IGE seems significant but lacks further validation or comparison with other studies in similar contexts

Prediction Accuracy:

While ssGBLUP improves prediction accuracy, the marginal gains (6.35% for HW, 10.53% for HWU) may not justify the additional cost and effort involved in genotyping.

The MSEP for ssGBLUP is higher than for pBLUP, suggesting potential biases or overfitting in the genomic model, which warrants further investigation.

In the conclusion section

The conclusions could be more critical and nuanced. For instance:

While the study shows that incorporating IGE improves model accuracy, it does not address the practical challenges (e.g., cost, computational complexity) of implementing these methods in commercial shrimp breeding programs.

The study emphasizes the feasibility of IGE in shrimp due to their large family sizes and short production cycles but does not discuss potential limitations, such as the difficulty in scaling this approach to broader populations.

Comments on the Quality of English Language

English Language, Good

Author Response

Comments 1: In the abstract section there are not need to mention abbreviation of short for the should be start from introduction.

Response 1: Thanks for your suggestions. We have revised the "Abstract" section as per your suggestions.

Comments 2: In the material method section, while the study highlights the advantage of using the 3FAM design for indirect genetic effects (IGE), it does not provide sufficient detail on how the cages were assigned or randomized. The lack of randomization procedures may introduce bias.

Response 2: Thank you for your suggestions. We are sorry that we didn’t state it clearly. Based on three parameters, a software was developed to allocate families to cages numbered 1 to 40:  the number of the same families in any two cages, the coefficient of variation of body weight within each cage, and the kinship coefficient between families within each cage (luan et al., 2020, Ødegård et al., 2011, and Sae-Lim et al., 2016). The allocation principle of the software is that any specific pairwise family combination appears in only one cage throughout the experiment. In each cage, the coefficient of variation for the three families and all individuals (both family and individual variations) must be less than or equal to the coefficient of variation for the total 40 families or all tested individuals. Additionally, the average kinship coefficient between the three families in each cage must be less than or equal to the average kinship coefficient among the 40 families. The relevant content has been added to the original text. (Lines 141-146) 

Reference:

Ødegård, J.; Olesen, I. Comparison of testing designs for genetic evaluation of social effects in aquaculture species. Aquaculture 2011317, 74-78.

Luan, S.; Luo, K.; Chai, Z.; Cao, B.; Meng, X.; Lu, X.; Liu, N.; Xu, S.; Kong, J. An analysis of indirect genetic effects on adult body weight of the Pacific white shrimp Litopenaeus vannamei at low rearing density. Genet Sel Evol. 2015, 47, 1-8.

Sae-Lim, P.; Bijma, P. Comparison of designs for estimating genetic parameters and obtaining response to selection for social interaction traits in aquaculture. Aquaculture 2016451, 330-339.

Comments 3: The trial duration (55 days) seems short, given the objectives to evaluate growth uniformity, which may benefit from longer observation periods to capture full genetic and environmental interactions.

Response 3: Thank you for your comment. You have raised a very valid point. The reason we chose a shorter mixed rearing test period is that approximately 60 days of individual rearing were required in the early stage to achieve the VIE marking specifications. In future studies, we plan to adopt a more flexible design, such as conducting mixed rearing tests at the P20 stage and using molecular marker techniques for pedigree reconstruction in the later stages. This approach will allow for an extended trial period to more comprehensively evaluate growth uniformity and its genetic and environmental interactions. Thank you again for your suggestion, which will greatly benefit the design of our future studies. 

Comments 4: In the Genotype and quality control, the study does not address potential limitations of using the 55K SNP Panel or whether higher-density panels would improve accuracy for such low-heritability traits.

Response 4: Thanks. For low-heritability traits of harvest weight uniformity, Sae-Lim et al demonstrated that the heritability of uniformity traits is 0.015 to 0.036, combining the animal-DHGLM model with ssGBLUP using 30,092 SNPs improved predictive ability by 41.1%–78.1% compared to pBLUP (Sae-Lim et al., 2017). Reproduction traits generally belong to the low heritability, Song et al reported that ssGBLUP produced 30 to approximately 38% higher accuracy than GBLUP, when the SNP density is 68K (Song et al., 2017). However, the prediction ability varies according to species, data structure and traits, etc. In the next step, it is necessary to verify the prediction ability of the model according to different marker densities. Thank you again for your good suggestion.

Reference:

Sae-Lim, P.; Kause, A.; Lillehammer, M.; Mulder, H.A. Estimation of breeding values for uniformity of growth in Atlantic salmon (Salmo salar) using pedigree relationships or single-step genomic evaluation. Genet Sel Evol. 2017, 49, 33.

Song, H.; Zhang, J.; Jiang, Y.; Gao, H.; Tang, S.; Mi, S.; ... Ding, X. Genomic prediction for growth and reproduction traits in pig using an admixed reference population. J Anim Sci. 2017, 95, 3415-3424.

Comments 5: In the statistical analysis, the study faced convergence issues in the models incorporating IGE (A_IGE and H_IGE), which were subsequently excluded from cross-validation. This limitation weakens the strength of conclusions drawn about IGE's influence.

Response 5: This is a very important issue. The convergence of the model is influenced by various factors, such as model complexity and data structure. This has indeed weakened the evidence for IGE to some extent. An investigation was performed on the IGE model without cage effects to further evaluate the importance of IGE. Specifically, a 10-fold cross-validation with 5 replicates was used to assess the impact of IGE (A_IGE vs. A_NoIGE) on the prediction accuracy of HW and HWU. The results showed that for HW, the prediction accuracy increased from 0.313 (A_NoIGE) to 0.371 (A_IGE), an improvement of 18.53%. For HWU, the prediction accuracy increased from 0.038 to 0.046, representing an increase of 21.05%. (Lines 383-391) 

Comments 6: the mathematical equations for variance components are provided but could benefit from a clearer explanation or visual representation for accessibility to a broader audience.

Response 6: Thank you for your suggestions. We have revised in the article (Lines 231). 

Comments 7: In the result section, the heritability estimates for HWU (0.005–0.017) are exceptionally low, and while the study acknowledges this, it does not sufficiently discuss potential reasons, such as environmental noise or measurement errors.

Response 7: Thank you for your comment. On the one hand, the low heritability estimates for HWU (0.005–0.017) is determined by the trait, which has been confirmed by many reports above.

On the other hand, low heritability estimates for harvest weight uniformity can be primarily attributed to the fact that individual dispersion (i.e., the magnitude of data variation) cannot be directly measured but must be inferred through residuals. Residuals represent the differences between observed values and expected values. Due to the nature of residuals, their mean typically approaches zero, and they are symmetrically distributed around zero regardless of the true variance. As a result, data from a single individual provides limited information for accurately estimating the underlying variance. However, when examining data across multiple families (e.g., different family groups), the differences between families become more pronounced, and these differences can be quantified through statistical methods, enabling a more effective estimation of variance. So, a larger number of family size can help with better assessment. Additionally, relevant studies also indicate that increasing repeated records allows the model to estimate at the individual level, thereby achieving higher heritability estimates. (Lines 338-343)

Comments 8: The large increase in HWU heritability (150–240%) after incorporating IGE seems significant but lacks further validation or comparison with other studies in similar contexts.

Response 8: Thank you for your comment. Currently, there is no relevant literature that has compared or validated such changes in similar contexts. Therefore, we have conducted a preliminary exploration in this study and hope to provide new perspectives and data for future research. Additionally, as mentioned earlier, the significant increase in accuracy when including IGE in the model, compared to the model without IGE, further supports the validity of the observed significant increase in HWU.

Comments 9: While ssGBLUP improves prediction accuracy, the marginal gains (6.35% for HW, 10.53% for HWU) may not justify the additional cost and effort involved in genotyping.

Response 9: Thank you for your feedback. We understand your concern. In practical production settings, it is feasible to reduce SNP density to lower genotyping costs while maintaining reasonable prediction accuracy. Genotype imputation is an effective strategy to further optimize costs. For instance, by leveraging 55K genotypic information from parents and some sibling individuals, the genotypes of test individuals can be imputed from 2-3K to 55K, significantly reducing the overall genotyping cost. Future studies will explore these optimization strategies to balance cost and prediction performance.

Comments 10: The MSEP for ssGBLUP is higher than for pBLUP, suggesting potential biases or overfitting in the genomic model, which warrants further investigation.

Response 10: Thank you for your comment. Firstly, we apologize for the writing errors in the article, the indicator parameter of the prediction bias should be “bias” instead of “MSEP”, in fact, the calculated results in the manuscript were “bias” of model. We have revised it in the whole article.  (Lines 258-260) 

Prediction accuracy typically measures the correlation between the predicted value and the true value, while prediction bias reflects a systematic shift (such as underestimation or overestimation) of the model's predicted value. GS uses genomic information to better capture true genetic differences between individuals and families, and thus may be better than traditional BLUP models in predicting accuracy. For the bias, there are several likely reasons for the more deviated regression from unity for ssGBLUP. Genotyped parents were not random samples from the whole population, but selected individuals, which may in part lead to the prediction bias. And, SNP markers and causal genes were not in complete linkage disequilibrium due to the small number of genotyped individuals, the large sampling variance and the limited marker density, and thus could not fully account for the additive genetic effect, which may have a larger prediction deviation due to overfitting. For milk yield of dairy cattle, the regressions for BayesB and GBLUP decreased from 1.77 to 1.03 and 1.38 to 1.04, respectively, when the number of genotyped increased from 125 to 2000 (Zhang et al., 2014).

In the genomic prediction study of fleece traits in Inner Mongolia Cashmere goats by Yan et al. (Yan et al., 2020), although the ssGBLUP method showed significantly higher accuracy than the ABLUP method (29% ~ 33%), the prediction bias of ssGBLUP (1.3182) was also higher than that of pBLUP (1.1865). (Lines 457-477)  

Reference:

Zhang, Z.; Ober, U.; Erbe, M.; Zhang, H.;Gao, N.; He, J.L.; Li, J.Q.; Simianer, H. Improving the accuracy of whole genome prediction for complex traits using the results of genome wide association studies. PLoS One 2014, 9, 3.

Yan, X.; Li, J.; He, L.; Chen, O.; Wang, N.; Wang, S.; et al. Accuracy of Genomic prediction for fleece traits in Inner Mongolia Cashmere goats. BMC genomics 2024, 25, 349.

Comments 11: In the conclusion section, while the study shows that incorporating IGE improves model accuracy, it does not address the practical challenges (e.g., cost, computational complexity) of implementing these methods in commercial shrimp breeding programs.

Response 11: Thank you for your valuable feedback. The 3FAM design requires strict control of family combinations, individual identification, and tracking, raising management costs and labor demands, which may limit its practicality. Moreover, incorporating IGE, while improving analytical accuracy, increases computational complexity. Therefore, it is necessary to optimize the design and model to reduce costs and improve feasibility.

Comments 12: The study emphasizes the feasibility of IGE in shrimp due to their large family sizes and short production cycles but does not discuss potential limitations, such as the difficulty in scaling this approach to broader populations. 

Response 12: Thank you for your comment. We have added a discussion on this point in the original manuscript. (Lines 397-403) 

Round 2

Reviewer 2 Report

Comments and Suggestions for Authors

Reviewer Report

Manuscript id: biology-3413836R1

Manuscript entitle: “Genetic Analysis of Harvest Weight Uniformity in Litopenaeus vannamei Incorporating Indirect Genetic Effects through Ge-nomic Information”

The revision is satisfactory, and the manuscript can be accepted.

Author Response

Comments 1: The revision is satisfactory, and the manuscript can be accepted.

Response 1: Thank you very much for your positive feedback. We are glad to hear that the revisions meet your expectations, and we appreciate your time and effort in reviewing our manuscript.